# OpenReview forum: "Self-Supervised Alignment with Mutual Information: Learning to Follow Principles without Preference Labels"
_NeurIPS.cc/2024/Conference — NeurIPS 2024 poster_

### Official Review · Reviewer_5Enh · 2024-06-28

**Soundness:** 3
**Presentation:** 4
**Contribution:** 3
**Rating:** 6
**Confidence:** 3

**Summary:**

The paper introduces SAMI, an iterative algorithm designed to align LLMs with behavioral principles (constitutions) without the need for preference labels or demonstrations. SAMI achieves this by optimizing the conditional mutual information between principles and self-generated responses given queries. The approach shows significant improvements in single-turn dialogue and summarization tasks compared to pretrained and instruction-finetuned models.

In summary, this paper presents a novel and effective method for aligning language models with behavioral principles without relying on preference labels or demonstrations. Despite demonstrating strong empirical results and scalability, the approach faces challenges related to its dependence on the initial models, regularization issues, domain limitations, and length bias. Overall, SAMI represents a significant advancement in the efficient and practical alignment of language models.

**Strengths:**

1. The introduction of SAMI represents a significant innovation by aligning LMs with principles without using preference labels or human demonstrations.
2. The method outperforms both the initial pretrained model and an instruction-finetuned baseline in single-turn dialogue and summarization tasks.
3. SAMI scales effectively to stronger models (e.g., llama3-70b) and generalizes to diverse principles not seen during training.
4. The ability to align LMs to follow principles without extensive human oversight has practical implications for reducing the resource intensity and complexity of current alignment techniques.

**Weaknesses:**

1. The approach faces potential over-optimization issues, producing non-coherent outputs (gibberish) if not regularized properly. The paper mentions that current regularization strategies add algorithmic complexity and are not always effective.

2. The experiments are restricted to single-turn dialogue and summarization tasks. It remains to be seen how SAMI performs on more complex multi-turn interactions and a broader range of tasks. Additionally, there is a noted length bias in responses, especially when using mutual information as a measure, which can affect the quality and coherence of the generated outputs.

3. The experimental results (such as Figure 2) suggest that multiple iterations are needed to achieve optimal performance, raising concerns about the potential resource burden. The authors should further elucidate this aspect. Additionally, it would be beneficial to include more ablation studies and comparisons with other alignment methods to provide a comprehensive evaluation of SAMI's effectiveness.

**Questions:**

1. Can the authors provide more details on the regularization strategies they have considered? What specific measures have they found effective in mitigating over-optimization and producing coherent outputs?
2. Have the authors considered extending their experiments to multi-turn interactions and more complex tasks? If so, what are the preliminary findings, if any?
3. The experimental results indicate that multiple iterations are needed to achieve optimal performance. Can the authors provide more insights into the resource requirements and computational costs associated with these iterations?

**Limitations:**

1. While the authors acknowledge the issue of over-optimization leading to gibberish outputs, a more detailed discussion on potential solutions and future directions for regularization would be helpful. This would provide readers with a clearer understanding of how to tackle this challenge.
2. The current experiments are limited to single-turn dialogue and summarization tasks. Expanding the discussion to include potential performance in multi-turn interactions and a wider range of tasks would address concerns about the generalizability of SAMI.
3. Including more ablation studies and comparisons with other alignment methods would strengthen the evaluation of SAMI. This would help readers understand its relative strengths and weaknesses better.

---

> ### Author Rebuttal · Authors · 2024-08-06
>
> Thank you for your positive evaluation of our work!
>
> As mentioned in our responses to our first reviewer LYCL, SAMI, like other alignment methods such as DPO, suffers from over-optimization (e.g., non-coherent outputs if trained for too long). We regularize against this “forgetting” by always starting from the initial model during finetuning, using data generated from an intermediate model. While this is a limitation, we do not add additional complexity through our regularization. In fact, by not having a reference model + KL divergence, we reduce algorithmic complexity as we only need to load one model into memory during training. We will make sure to better clarify this relationship and flag the over-optimization issue more clearly in our limitations.
>
> Regarding your other questions and concerns:
> - We agree that evaluation on more complex domains is an important limitation and plan to address this in future work.
>
> - Regarding length bias concerns: While HH-RLHF suffered from length bias, our results on TL;DR have shown that length bias can be regularized against simply by stating that responses should be concise. Fig. 3 shows that sequence lengths decreased over iterations as a result of including a conciseness principle in the constitution. As such, while the original objective suffers from a length bias, this can be regularized against by including conciseness as a part of the constitution. Moreover, we have now included updated results with a more principled length correction for our HH-RLHF experiments (see response to reviewer eqUp).
>
> - As mentioned in our response to reviewer LYCL, we only need a small amount of data at each iteration, which is why the computational/resource requirements for training a model are rather low. We will include ablations with respect to principles and potential simplifications of our objective in our revision.
>
> - Regarding regularization: Please see our response to LYCL.
>
> - We have not yet extended to multi-turn interactions; however, related work has looked at multi-turn interactions using self-improvement techniques and shown promising results (Andukuri et al., 2024). In future extensions, we plan to combine ideas from this multi-turn setting with our mutual information objective.
>
> References
> - Zelikman, E., Wu, Y., Mu, J., & Goodman, N. (2022). Star: Bootstrapping reasoning with reasoning. Advances in Neural Information Processing Systems, 35, 15476-15488.
> - Andukuri, C., Fränken, J. P., Gerstenberg, T., & Goodman, N. D. (2024). Star-gate: Teaching language models to ask clarifying questions. Conference o

---

> > ### Comment · Reviewer_5Enh · 2024-08-10
> > **Official Comment by Reviewer 5Enh**
> >
> > Thank you for the rebuttal. The authors have addressed most of my concerns. I am pleased that the computational/resource requirements for training the model are relatively low. I look forward to seeing the algorithms extended to multi-turn interaction settings. I will maintain my current score.

---

### Official Review · Reviewer_mS35 · 2024-07-06

**Soundness:** 3
**Presentation:** 3
**Contribution:** 3
**Rating:** 6
**Confidence:** 4

**Summary:**

This work presents a method to align the model towards a set of principles. The general idea is to sample responses from the model with different constitutions and optimize the matching between responses and constitutions via an infoNCE-type contrastive loss. The whole process is done iteratively to improve the alignment. This method is tested on single-turn dialog and summarization, and is shown to improve the performance of mixtral and llama3 models with constitutions sampled either from weaker or stronger models. Further experiments also demonstrate this method's capability to use diverse principles and to combine chain-of-thought reasoning.

**Strengths:**

1. This paper is well-written and easy to follow. All the detailed hyperparameters are listed in the appendix.

2. The improvements are good. It is nice to see the models can even benefit from principles generated by weaker models.

**Weaknesses:**

1. While the improvements are convincing, it would make this paper much stronger if the model could be compared with baselines. Some valuable baselines to compare with include: (1) simplified versions of the proposed method (e.g., simplifying the contrastive loss part); (2) baselines mentioned in the related work part (which I acknowledge that may not using the same resources, but would still be good to have).

**Questions:**

1. One of the motivations behind this work seems to be to remove the dependency on "carefully curated" examples. I'm wondering how robust this method is towards different principles. It would be great to show simple experiments on this or even just impressions through all the existing experiments.

2. I don't understand "by using a small number of gradient updates during earlier iterations" in line 145. How does this regularize distribution shift?

3. While the MI lower bound is increasing smoothly during training, the win rates are not. How do you determine the total number of iterations and select the best checkpoint during training?

**Limitations:**

Sec. 5 includes a paragraph discussing the limitations of this work.

---

> ### Author Rebuttal · Authors · 2024-08-06
>
> Thank you for your positive evaluation of our work. We completely agree that comparing to other baselines than instruct-finetuned models and base models directly is an important limitation of our work and we plan to address this in future extensions.
>
> Regarding your specific questions:
> - We agree that re-running experiments using simplified versions of our contrastive loss (e.g., one-sided vs. two-sided) are important ablations. We will include these in our revision.
> - We agree that our first two experiments with HH-RLHF and TL;DR only involved a small number of carefully curated examples. As such, our third experiment with Llama3-70B involves a larger selection of diverse principles (e.g., talk like a pirate or use emojis; see Fig. 5 and Section A.13). We provide examples in Section A.9 (see e.g., at the bottom of p. 23)
> - As mentioned in our responses to other reviewers, we are planning to evaluate additional domains requiring more diverse principles, such as roleplaying personas directly (e.g., in MT-bench) in future versions.
> - Regarding your comment "I don't understand 'by using a small number of gradient updates during earlier iterations' in line 145. How does this regularize distribution shift?": Thank you for pointing this out. We will revise this statement to be more precise in our revision. Specifically, we regularize in two ways: First, we always train the base model (i.e., the initial model) using data generated from each intermediate model. As such, the same model is never trained twice, and an intermediate model is always using new data for training the next iteration of a model. Second, by only taking a small number of gradient steps, we stay as close as possible to the initial model to avoid forgetting (see also our third response to reviewer LYCL).
> - Regarding checkpoint selection: Following previous work (Zelikman et al., 2022), we start with a small number of examples during the first iteration and linearly increase the number of training examples at each iteration. We fix both the number of iterations and the number of examples in advance (see also our third response to reviewer LYCL). We chose 3 iterations simply because we have to run GPT-win rates evaluations to get win rates after each iteration and need to generate new data, so each iteration is resource-intensive. Since related works (e.g., Andukuri et al., 2024) have observed a ceiling after or around three iterations, we followed this approach. We will make sure to point this limitation out more carefully in our revision.
>
> References:
> - Zelikman, E., Wu, Y., Mu, J., & Goodman, N. (2022). Star: Bootstrapping reasoning with reasoning. Advances in Neural Information Processing Systems, 35, 15476-15488.
> - Andukuri, C., Fränken, J. P., Gerstenberg, T., & Goodman, N. D. (2024). Star-gate: Teaching language models to ask clarifying questions. Conference on Language Modeling Research.

---

> > ### Comment · Reviewer_mS35 · 2024-08-12
> >
> > Thank you for the response! The explanations and additional details resolved most of my concerns, so I increased my score.

---

### Official Review · Reviewer_eqUp · 2024-07-12

**Soundness:** 3
**Presentation:** 3
**Contribution:** 3
**Rating:** 6
**Confidence:** 4

**Summary:**

The authors propose a technique to improve the ability of language models (LM) to abide by constitutions without using human labels. First, they ask a principle writer LM to construct detailed constitutions and inverse versions of them (called “antitheses” in this work). Then the main LM, which is the LM to be improved, is asked to generate a response to a prompt for each of the constitutions. Finally, a mutual information loss is calculated and backpropagated, which simultaneously encourages the LM to produce response y_1 under constitution c_1 and and to produce y_2 under c_2, while also discouraging the model to produce y_1 under c_2 and to produce y_2 under c_1 (where y_i was indeed produced when conditioned on c_i). The authors show strong improvements over baselines with this technique.

**Strengths:**

* The authors propose a novel technique for improving the steerability ability of LMs
* The improvement over baselines are impressive
* The paper is comprehensive in its results and analysis, and the Appendix is detailed
* The paper is well-written and easy to follow for the most part

**Weaknesses:**

* Soundness
  * I believe the length correction method is insufficiently rigorous, as it doesn’t account for the magnitude of length variations. Imagine a hypothetical scenario where win rate is entirely correlated to length. If your less-than-or-equal bucket is on average 100% of the base length, then it will achieve a win rate of 50%. Then say your greater-than bucket is 400% the base length, leading it to achieve 100% win rate. Averaging out the two win rates will yield a result of 75%, when in fact an unbiased length-corrected result should yield 50%. I recommend a more principled approach, like the one mentioned in Stiennon et al 2020 Appendix F - https://arxiv.org/abs/2009.01325, which uses logistic regression to remove the effect of length.
  * Please include statistical significance of win rates and other key results
  * Please include details on checkpoint selection and the train/validation/test splits used. From reading the paper alone, I am under the impression that there were no validation splits used, which would be concerning
* Usefulness of principle writer
  * Is the principle writer component necessary? It seems like it’s not a vital part of this technique, and a human could write a constitution given that they are already writing out specifications to the principle writer. It also takes up a lot of space in this paper and is rather distracting
  * From eyeballing Figs 3-4, it also looks like principle writer size doesn’t matter. This should be stated clearly in the results.
  * Ablation should be conducted on whether you need antitheses vs. just need more than 1 constitution. This fits nicely into the contrastive loss narrative, but it’s not clear if antitheses are really necessary.
* Clarity
  * It should be made clearer that the idea is to improve the steerability of the LLM via constitutions. When reading the paper, I was originally under the impression that the goal was to align the model to a specific constitution (e.g. “be helpful and harmless”).

**Questions:**

* Can you clarify whether the inputs to the SAMI and the baseline models are exactly the same? (both in the paper and in your response) This is quite important to making sure the authors are using a fair baseline
* Figures 2-4 are very busy and hard to interpret. Please consider trimming them down (e.g. removing the “principle writer size” dimension), or using different colors, symbols

**Limitations:**

Adequately addressed

---

> ### Author Rebuttal · Authors · 2024-08-06
>
> Thank you for your positive evaluation of our work and providing the additional length correction reference. We will make sure to report both statistical significance in addition to confidence intervals in our revision. Moreover, we will revise figures for clarity as requested.
>
> Please see our attached pdf for updated win rates from Experiment 1 (HH-RLHF Dialogue) based on fitting a logistic regression model to remove length effects. For fitting the logistic model, we have followed the most recent evaluation standard from Length-Controlled Alpaca Evals (Dubois et al., 2024) which similar to Stiennon et al. (2022), inputs the length of each response as well as the result and trains a classifier to predict the counterfactual: “what would the preference be if the model’s output had the same length as the baseline”. Results for HH-RLHF after applying this length correction show the same pattern as our previous results albeit being more conservative. Thank you again for pointing this out, we will make sure to update our figures and text to reflect this change!
>
> Regarding your other concerns and specific questions:
> - We take only one gradient step on each batch and never train twice on a given data point within a given iteration. Moreover, we alternate between two dataset splits (A, B) between iterations, ensuring that if a model was trained on split A it never sees data points from split A when generating new training data for the next iteration, for which we use split B (and vice versa if a model was trained on split A). As such, at every point in our pipeline (data generation, training, and win-rate evaluation), a data point encountered by a given model is seen for the first time and is never seen again. We then fix the number of iterations across experiments and compute win rates at the end of each iteration to evaluate performance (similar to Zelikman et al., 2022).
> - While it is necessary to have principles, the principle writer could be both a human or another language model. Moreover, principles could be sampled from a pre-existing dataset. As such, the principle writer is not strictly necessary for our pipeline. However, we would like to emphasize that we were interested in exploring a setting in which a principle writer might be weaker than the student being finetuned. We believe that—similar to using a weak supervisor model to label data for training a strong student (i.e., weak-to-strong generalization; see e.g., Burns et al.; 2023)—this is an important point to make on its own as it is not unlikely that future models might surpass human users in capabilities while still having to follow instructions human users. As you mention in your next comment, Figures 3-4 show that a small principle writer can indeed be used to align a stronger student, which we believe is a key finding suggesting that small aligned principle writers / models (which act as a stand-in for a human user) can be used to steer strong students. Thank you for pointing this out again, we will make sure to state this more clearly in our results!
> - We agree that additional ablations for the usefulness of antitheses are important and will include these future extensions of our work.
> - We agree that we need to make our goal—steerability of a language model via constitutions—more explicit. This goal is distinct from aligning a model to a specific constitution (or a specific distribution of labels through RLHF). Instead, it aims to increase steerability more generally.
> - Yes, the inputs to SAMI and baselines are the same. Both the original base model and SAMI-finetuned models use a base model template with no additional special tokens except BOS and EOS tokens. Instruct-model based templates use the exact same input, with the only difference being the additional special tokens required by the tokenizer.
>
> References:
> - Burns, C., Izmailov, P., Kirchner, J. H., Baker, B., Gao, L., Aschenbrenner, L., ... & Wu, J. (2023). Weak-to-strong generalization: Eliciting strong capabilities with weak supervision. arXiv preprint arXiv:2312.09390.
>
> - Dubois, Y., Galambosi, B., Liang, P., & Hashimoto, T. B. (2024). Length-controlled alpacaeval: A simple way to debias automatic evaluators. arXiv preprint arXiv:2404.04475.
>
> - Zelikman, E., Wu, Y., Mu, J., & Goodman, N. (2022). Star: Bootstrapping reasoning with reasoning. Advances in Neural Information Processing Systems, 35, 15476-15488.

---

> > ### Comment · Reviewer_eqUp · 2024-08-09
> >
> > Thank you for addressing my concerns, especially regarding the length analysis. Please include significance testing in your next revision as promised. I will raise my score accordingly.

---

### Official Review · Reviewer_LYCL · 2024-07-17

**Soundness:** 3
**Presentation:** 4
**Contribution:** 3
**Rating:** 6
**Confidence:** 4

**Summary:**

This paper introduces SAMI (Self-Supervised Alignment with Mutual Information), an iterative algorithm for aligning language models to follow behavioral principles without using preference labels or demonstrations. The key idea is to finetune a pretrained LM to increase the mutual information between constitutions (sets of principles) and self-generated responses. The authors demonstrate SAMI's effectiveness on dialogue and summarization tasks, showing it can outperform both the base model and instruction-tuned baselines. They also show SAMI can align strong models using principles written by weaker models, and that it generalizes to diverse principles and scales to larger models.

**Strengths:**

1. The paper presents a novel approach to language model alignment that does not require preference labels or demonstrations. This is a significant departure from existing methods like RLHF or supervised finetuning.
2. The method is well-developed and grounded in information theory. The authors provide a clear theoretical foundation for their approach, deriving a tractable lower bound on the conditional mutual information objective.
3. The paper is well-structured and clearly written. The SAMI algorithm is presented in detail (Algorithm 1), and the key ideas are illustrated effectively through Figure 1. If the results hold up to scrutiny, this could be an important contribution to the field of AI alignment. The ability to align language models without relying on expensive and potentially biased human preference data could significantly accelerate progress in this area.
4. Authors conduct a comprehensive set of experiments, including comparisons to strong baselines, investigations of generalization to diverse principles, and scaling to larger models (llama3-70b).

**Weaknesses:**

1. While the paper shows results on dialogue and summarization tasks, it would be beneficial to see performance on a wider range of tasks to better understand the method's generalizability.
2. The paper compares primarily to instruction-tuned models and base models. Comparisons to more recent alignment methods like constitutional AI or RLAIF would strengthen the results.
3. While the authors mention regularization to prevent divergence from the initial model, there's limited discussion of how this affects the model's ability to generalize to new tasks or domains not seen during training.
4. While the method is shown to work with llama3-70b, it's not clear how computationally intensive the approach is compared to other alignment methods, especially for very large models.

**Questions:**

1. How does the computational cost of SAMI compare to other alignment methods like RLHF or constitutional AI, especially for very large models?
2. Have you investigated how SAMI performs on tasks significantly different from those used in training? For example, if trained on summarization, how well does it generalize to tasks like code generation or mathematical reasoning?
3. The paper mentions using regularization to prevent divergence from the initial model. Can you provide more details on how this regularization affects the model's ability to learn new behaviors not present in the initial model?

**Limitations:**

None.

---

> ### Author Rebuttal · Authors · 2024-08-06
>
> Thank you for your positive evaluation of our work!
>
> We completely agree that future work should include a wider range of tasks. Widening both the range of tasks and principles is crucial for training a general constitution-following model, and we plan to address this limitation in future extensions of our work. We will ensure that this limitation is carefully addressed in our limitations section.
>
> Regarding your questions:
> - SAMI's computational cost is relatively low. We apply only one gradient step per batch, with a batch size of 128. At each iteration, we train on at most 12 batches (1,536 examples). Following Zelikman et al. (2022), we start with a small number of batches and increase the number of batches by a constant factor in later iterations. Consequently, fine-tuning small models like Mistral-7B or Mixtral-8x7B takes no more than 5-10 minutes at a given iteration (using reasonable hardware such as A100 GPUs). While fine-tuning LLaMA-70B is more expensive due to its size, the dataset still remains small (≤1,536 examples at each iteration; see section A.3 for further details). For comparison, the original DPO paper used 170,000 HH-RLHF examples (see p. 7 in Rafailov et al., 2023), an additional SFT stage prior to DPO fine-tuning, and trained for multiple epochs. Importantly, we view SAMI as a complementary approach to other alignment fine-tuning methods, not a replacement. SAMI's goal is to enhance a model's steerability by amplifying the connection between a set of guiding principles and the responses that realize them, and it can in principle be applied at later stages in post-training, e.g., after DPO or instruction finetuning.
>
> - As mentioned above, we have not yet evaluated SAMI on other domains. This is an important limitation that we will carefully address in our limitations section. An important aspect to consider is that training on summarization and evaluating on mathematical reasoning is unlikely to benefit a model. Instead, we anticipate that training on a variety of domains should help the model generalize. For example, jointly training on code generation, mathematical reasoning, summarization, and other domains should increase generalization performance. We note that this limitation is not specific to SAMI but a data limitation that should apply to finetuning/alignment methods more generally.
>
> - To clarify, we regularize by training the initial (i.e., original base) model at each iteration, using data generated from each intermediate model. This approach is standard (see Zelikman et al., 2023; Andukuri et al., 2024) and prevents overfitting. We only train on a small number of examples using a low learning rate to avoid substantial changes to the initial model. The reason for using regularization is that, as with other alignment approaches like RLHF and DPO which require a reference model and KL divergence, a model might exploit the objective to obtain more reward. In our case, this could mean pushing the log probabilities to an identity matrix to maximize mutual information. Thus, regularization does not prevent the model's ability to learn new behaviors but instead prevents it from "forgetting" desirable behaviors due to reward overoptimization.
>
> References:
> - Zelikman, E., Wu, Y., Mu, J., & Goodman, N. (2022). Star: Bootstrapping reasoning with reasoning. Advances in Neural Information Processing Systems, 35, 15476-15488.
> - Andukuri, C., Fränken, J. P., Gerstenberg, T., & Goodman, N. D. (2024). Star-gate: Teaching language models to ask clarifying questions. Conference on Language Modeling Research.
> - Rafailov, R., Sharma, A., Mitchell, E., Manning, C. D., Ermon, S., & Finn, C. (2023). Direct preference optimization: Your language model is secretly a reward model. Advances in Neural Information Processing Systems, 36.

---

> > ### Comment · Reviewer_LYCL · 2024-08-12
> >
> > Thanks the authors for their response. After reading the response, I think my current score is appropriate.

---

### Author Rebuttal · Authors · 2024-08-06

We would like to thank the reviewers for their positive evaluation of our work and their helpful suggestions for revising our paper.

We have included length-corrected dialogue win rates based on logistic regression (as requested by reviewer eqUp) in the attached .pdf.

Point-by-point responses to each reviewer are provided below.

---

### Decision · Program_Chairs · 2024-09-25

**Decision:**

Accept (poster)

**Comment:**

This paper proposes a new, iterative algorithm, SAMI, that fine-tunes an LM with the goal of increasing the conditional mutual information between constitutions and self-generated responses given queries from a dataset, hence without requiring any preference labels or demonstrations. Reviews are in general positive and the rebuttal includes clarifications and additional information responding to the weaknesses and suggestions.

Authors can include significance testing in the paper, as suggested by one of the reviewers and as promised in the rebuttal.